# Nanomedicine Strategies in the Management of Inflammatory Bowel Disease and Colorectal Cancer

**DOI:** 10.3390/ijms26136465

**Published:** 2025-07-04

**Authors:** Asia Xiao Xuan Tan, Brandon Yen Chow Ong, Tarini Dinesh, Dinesh Kumar Srinivasan

**Affiliations:** 1Yong Loo Lin School of Medicine, National University of Singapore, Singapore 117597, Singapore; e1177525@u.nus.edu (A.X.X.T.); e1237595@u.nus.edu (B.Y.C.O.); 2Department of Anaesthesia, Yong Loo Lin School of Medicine, National University of Singapore, Singapore 119074, Singapore; tarinidinesh@gmail.com; 3Department of Anatomy, Yong Loo Lin School of Medicine, National University of Singapore, Singapore 117594, Singapore

**Keywords:** gut dysbiosis, inflammatory bowel disease, colorectal cancer, nanomedicine, nanodiagnostics, nanotherapeutics, drug delivery systems

## Abstract

The gut microbiota has emerged as a key area of biomedical research due to its integral role in maintaining host health and its involvement in the pathogenesis of many systemic diseases. Growing evidence supports the notion that gut dysbiosis contributes significantly to diseases and their progression. An example would be inflammatory bowel disease (IBD), a group of conditions that cause inflammation and swelling of the digestive tract, with the principal types being ulcerative colitis (UC) and Crohn’s disease (CD). Another notable disease with significant association to gut dysbiosis would be colorectal cancer (CRC), a malignancy which typically begins as polyps in the colon or rectum, but has the potential to metastasise to other parts of the body, including the liver and lungs, among others. Concurrently, advances in nanomedicine, an evolving field that applies nanotechnology for disease prevention, diagnosis, and treatment, have opened new avenues for targeted and efficient therapeutic strategies. In this paper, we provide an overview of the gut microbiota and the implications of its dysregulation in human disease. We then review the emerging nanotechnology-based approaches for both therapeutic and diagnostic purposes, with a particular focus on their applications in IBD and CRC.

## 1. Introduction

### 1.1. The Gut Microbiota

The gut microbiome is a complex and dynamic system that originates with vertical transmission from the mother in the uterus, with full-term infants developing extensively colonised gut microbiomes [1]. The gut receives a constant stream of microbes, which compete for resources and evade the host’s immune system to colonise the gut [2]. It is colonised by humongous populations of trillions of microorganisms, in which *Bacteroidetes*, *Firmicutes*, *Proteobacteria*, *Fusobacteria*, *Tenericutes*, *Actinobacteria*, and *Verrucomicrobia* dominate [3]. The altering gut microbiome has great inter-person variability and undergoes extensive changes with age [4]. Many factors play a role in affecting its composition, including the host’s diet [5], genetics, and immune response [6]. The idea of a healthy gut microbiome is thus hard to define, with possible definitions including “a stable microbiome community that can resist environmental stresses” or “a state where important metabolic pathways are preserved” [7].

The gut microbiome plays an essential role in maintaining the health of the gastrointestinal tract (GIT). It maintains a thick mucus layer to prevent colonisation of the gut by foreign pathogens [8] and competes for both nutrients and attachment sites on the intestinal walls, preventing pathogen invasion [2,9]. It also promotes the production of antimicrobial molecules [10]. Another key role of the gut microbiome is immunomodulation, as the gut microbiome presents a vast number of antigens to the immune cells of the mucosa. A fine balance must be maintained between tolerating the gut microbiome and controlling bacterial microbiome overgrowth [2]. Apart from regulating the local immune response, the gut microbiome also influences the host’s systemic immune responses [10]. The gut is the largest reservoir from which the immune system actively samples antigens, and its microbiome is suggested to influence the initiation, progression, and severity of autoimmune diseases, including rheumatoid arthritis [11] (Figure 1).

### 1.2. Gut Dysbiosis

Gut dysbiosis has been described as the disruption of the symbiotic balance between the microbiota and host [12]. It is caused by frequent changes to the environment or host related factors [13] that overpower the resistance and protective capabilities of the microbial system in the gut [14]. The host’s diet and antibiotic consumption are two major factors that contribute to gut dysbiosis. The gut microbiome is greatly affected by long-term dietary changes, with individuals from different communities having very different gut microbiomes [15]. Different proportions of carbohydrates, proteins, fats, food additives, and cooking methods exert different effects on the gut microbiome [5]. For example, animal-based diets result in a gut microbiome with more bile-tolerant bacteria such as *Bacteroides, Alistipes,* and *Biophilia* [16,17], but decreases plant polysaccharide-fermenting Firmicutes including *Roseburia*, *Eubacterium rectale*, and *Ruminococcus bromii* [16]. Acute changes in diet can also affect the gut microbiome; plain plant-based or animal-based diets followed for five days have a significant impact on microbial community structures [16]. Pathogen exposure is also a trigger of gut dysbiosis. In particular, enteric pathogens have demonstrated a high potential to cause microbial dysbiosis in mouse models [18]. Similarly, foodborne viruses induce both local and systemic inflammation that alter microbial composition [19]. Dysbiosis can promote unfavourable microbiota proliferation that impairs the integrity of the intestinal barrier [20]. This facilitates the translocation of hazardous chemicals and promotes systemic inflammation, contributing to the pathogenesis of inflammatory bowel disease (IBD) [21].

The gut microbiome is also severely impacted by the administration of antibiotics, where broad-spectrum antibiotics can affect the abundance of 30% of the bacteria in the gut community, causing a marked decrease in taxonomic diversity and richness [22,23]. Antibiotics also alter the gut microbiota’s gene expression, protein activity, and overall metabolism [24,25]. For instance, it has been found that the use of beta-lactams in a host alters carbohydrate degradation and sugar metabolism to mimic those of an obese individual [26]. These antibiotic-induced microbiota alterations can remain after long periods of time, ranging from months to years [27]. Notably, another study revealed that mice who were administered antibiotics suffered gut dysbiosis that eventually resulted in cognitive impairment [28]. Other causative factors of gut dysbiosis include host factors such as stress or infections [29].

## 2. Gut Dysbiosis in Pathogenesis of Diseases

Previous studies have shown that gut dysbiosis has profound implications on health and contributes to the pathogenesis of a range of diseases.

### 2.1. Gut Dysbiosis in Inflammatory Bowel Disease (IBD)

Gut dysbiosis has been found to play a key role in the pathogenesis of IBD [30], with patients exhibiting reductions in microbial population, functional diversity, and microbiome stability [31]. The differing gut microbiota play a key role in altering the metabolic pathways in the gut–although IBD patients only have a 2% difference in genus level clades, their metabolic pathways have a 12% difference when compared to healthy individuals [32]. The expression of genes related to oxidative stress and sulfate transport was found to be increased in IBD patients [32]. Processes including bile acid modification and short chain fatty acid (SCFA) modification are also altered in the intestinal microbiome of IBD patients [33,34,35] (Figure 1). These findings support that gut dysbiosis plays a role in the pathogenesis of IBD and is not merely an observed state as a result of the disease.

### 2.2. Gut Dysbiosis in Colorectal Cancer (CRC)

Changes in the gut microbiome also participate in the development of a substantial number of GIT malignancies [36,37], with links between gut dysbiosis and colorectal cancer (CRC) being found [38,39]. One mechanism involves the gut microbiome inducing genotoxic stress or producing metabolites that promote genetic and epigenetic alterations conducive to carcinogenesis [40]. Another pathway includes the modulation of SCFA levels, which can influence global chromatin architecture and subsequently affect gene expression and transcriptional regulation [41]. Butyrate, an SCFA, can modulate the apoptosis, proliferation, and invasion of several cancer cell lines, facilitating aberrant hyperproliferation of colon epithelial cells in mice [42]. The overgrowth of specific bacteria also plays a role in CRC development. *Fusobacterium nucleatum* directly promotes carcinogenesis when it secretes adhesin FadA that binds to the extracellular domain of E-cadherin on epithelial cells. The adhesin complex is dysfunctional and unable to bind to beta-catenin, causing beta-catenin to translocate to the nucleus. This upregulates mitogenic signalling, thereby increasing the expression of transcription factors and stimulating the growth of CRC cells [43]. *Bacteroides fragilis* secretes the *B. fragilis* toxin, which cleaves E-cadherin, a tumour suppressor protein. This results in enhanced nuclear Wnt/b-catenin signalling that increases colonic carcinoma cell proliferation and the expression of the protooncogene MYC [44] (Figure 1).

## 3. Current Therapeutic Strategies

### 3.1. IBD Treatment

The conventional pharmacological therapy for IBD primarily aims to achieve and sustain remission and alleviate the secondary symptoms of the disease, rather than to correct the root pathogenic mechanism [45,46]. Commonly used drugs include corticosteroids, aminosalicylates, and immunosuppressive agents [47], while metronidazole and broad-spectrum antibiotics may be found useful at times [46]. Specific drugs are preferred for various aims. Steroids are selected for the short-term control of moderate to severe flare-ups but are unsuitable for long-term use due to their many systemic adverse effects and inability to maintain remission. At the same time, immunosuppressants, such as azathioprine, are preferentially employed in long-term therapy due to their delayed onset of therapeutic action [46].

Patients suffering from mild to moderate IBD are first treated with 5-aminosalicylic acid (5-ASA) before being given systemic glucocorticosteroids for more severe disease [48]. Azo-linked prodrugs of 5-ASA include sulfasalazine, olsalazine, and balsalazide, which are designed to deliver 5-ASA (mesalamine) specifically to the colon. These prodrugs remain intact through the upper GIT before being activated in the colon by azoreductase enzymes produced by colonic bacteria, cleaving the azo bond and releasing 5-ASA [46]. 5-ASA exerts its anti-inflammatory effects via the activation of peroxisome proliferator-activated receptor-gamma (PPAR-γ), a nuclear receptor highly expressed in colonic epithelial cells, whose signalling is otherwise impaired in IBD [49]. Though 5-ASA is generally well tolerated, the most common adverse effects include diarrhoea, nausea/vomiting, headache, abdominal pain/dyspepsia, rash, fever, fatigue/weakness, and arthralgia/myalgia [50].

The preferred corticosteroid therapy for IBD is prednisolone, whose administration can be done orally, rectally, or in urgent emergencies, parenterally [46]. Glucocorticoids modulate immune responses by binding to glucocorticoid receptors, regulating pro-inflammatory genes and suppressing immune-controlling transcription factors like nuclear factor κB (NF-κB). They also exert rapid non-genomic effects on cell signalling and ultimately reduce immune cell activation and pro-inflammatory cytokine production and promote apoptosis in dendritic cells and T cells [48]. However, the numerous adverse effects of corticosteroids remain a concern, with long-term use impacting patients’ metabolic and endocrine functions. These include weight gain, changes of fat distribution, hyperglycemia leading to diabetes, hypertension, reduced muscle and bone mass leading to osteoporosis and fractures, changes to skin with acne, cataracts, glaucoma, and depression [51]. There is also a pronounced risk of opportunistic infections when corticosteroids are prescribed concomitantly with other immunosuppressive drugs or biologics [52].

In patients with severe IBD or who are already steroid-resistant or dependent, immunosuppressants such as azathioprine and thiopurine derivative mercaptopurine are being used [53]. Azathioprine is a prodrug that metabolises into mercaptopurine, which is further converted into 6-thioguanine nucleotides—the primary active compounds that suppress purine synthesis and inhibit cellular proliferation [46]. However, the use of azathioprine and 6-mercaptopurine is limited by prominent adverse effects, including allergic reactions such as fever or rash or both, arthritis, leukopenia, pancreatitis, and nausea. In a study evaluating their efficacy in reducing remission in active Crohn’s disease, 27 out of 302 patients withdrew due to severe adverse events, highlighting the safety concerns of these immunosuppressants as IBD treatment [53].

Anti-tumour necrosis factor-alpha (anti-TNF-α) monoclonal antibodies such as Infliximab can induce apoptosis in mucosal T cells [54], macrophages, and monocytes [55]. These antibodies are also able to modulate fibroblast function [56], decrease leukocyte migration [57], and induce mucosal regulatory macrophages [58]. However, the development of tolerance to these drugs [59] and the increased frequency of serious infections are the downsides to this treatment [60]. Another antibody used is ustekinumab, a fully humanised antibody that blocks the p40 subunit of interleukin (IL)-12 and IL-23, showing consistent benefits over the placebo group when treating ulcerative colitis (UC) and Crohn’s disease (CD) as both induction and maintenance therapy [61,62]. The rate of antidrug antibodies was found to be low, with the rates of adverse events occurring being similar between the group given ustekinumab and the placebo group [61].

Alternative therapies also include cholestyramine, sodium cromoglycate, bismuth, arsenical salts, methotrexate, and fish oils [46]. Though primarily established as a treatment methodology for recurrent and refractory *Clostridium difficile* infections, faecal microbiota transplantation (FMT) is also potentially indicated for IBD [63]. Since its recognition by the United States Food and Drug Administration (FDA) in 2013, the range of FMT applications have extended beyond *Clostridium difficile* infections. They include other gastrointestinal diseases such as IBD and irritable bowel syndrome, and even extra-gastrointestinal diseases including obesity, metabolic syndrome, and severe multiple sclerosis [63]. It is essentially stool transplantation, where stools from a healthy donor are transferred into another patient’s GIT to directly compensate for the gut dysbiosis present, and FMT via multiple infusions administered via the lower GIT appears very promising for IBD treatment [64]. However, thorough and cautious processes of faecal and recipient preparation are required for a successful FMT [63].

### 3.2. CRC Treatment

Pre-cancerous and early-stage CRC can be removed by an endoscope, a safer and less invasive procedure than surgery [65]. For more advanced stages of CRC with liver metastases, surgical resection remains the only cure [66]. Lymph node dissection is also performed in surgery for staging and to determine the patient’s prognosis [65]. While surgery is an important first line of defence, surgery has been shown to increase the mortality risk in some cancer patients by metastasis, as it disrupts the integrity of the tumour [67]. Chemotherapy is used to treat CRC as well and can be used for neoadjuvant therapy, adjuvant chemotherapy, or chemotherapy for unresectable CRC [65]. Chemotherapeutic agents include cytotoxic drugs, molecular targeted drugs, and immune checkpoint inhibitors [65]. Traditional chemotherapy agents can also cause systemic effects such as nausea, vomiting, diarrhoea, and neuropathy due to their effects on other rapidly dividing cells in the body [67].

Over the past few decades, patients with CRC were treated homogeneously, with the same standard chemotherapy drugs administered after surgical resection [68]. Precision medicine is set to change that, where factors including environmental, lifestyle, cancer staging, and biological characteristics will be evaluated to identify the most beneficial treatment approach. This is done in hopes of improving treatment response and reducing the likelihood of side effects [69]. Drugs such as anti-vascular endothelial growth factor (VEGF) and anti-epidermal growth factor receptor (EGFR) monoclonal antibodies [68] allow for a more targeted treatment with decreased systemic side effects. While the use of such drugs can avoid systemic side effects, they can potentially result in a rash appearing on the upper body [67]. Radiotherapy is also used to treat locally advanced rectal cancer as an adjuvant to therapy, or as palliative care to preserve the patient’s quality of life [65].

## 4. Nanomedicine

Nanomedicine is an up-and-coming technology and branch of evolving medicine that uses nanotechnology for disease prevention, monitoring, and intervention through new modalities for the imaging, diagnosis, treatment, repair, and regeneration of biological systems [70]. Nanoparticles have unique chemical and physical properties that make them suitable for optical imaging, allowing for highly sensitive diagnostic tools with reduced degrees of invasiveness [71]. The high surface area to volume ratio of nanoparticles, combined with their ability to have surface modifications, enable them to deliver therapeutic agents to target sites more accurately, reducing unwanted side effects and thereby improving treatment efficacy [71]. Nanomedicine has revealed a promising future, with third-generation nanovectors bearing multi-functionality. Externally, there are recognition units for nanoparticles to interact with target sites, while internally, they are drug-loaded to deliver their desired therapeutic effect at the right place and time [72]. With its rapid development, nanomedicine has revealed great potential in becoming an asset to the healthcare industry by being able to overcome the limitations of traditional treatment methods. The different types of nanomedicine available and their uses in treating gut dysbiosis and its associated diseases will be discussed below.

## 5. Diagnostic Nanomedicine

### 5.1. Nanomedicine for Measuring Diagnostic Biomarkers

#### 5.1.1. Biomarkers of Gut Dysbiosis

Nanomedicine has proven its usefulness in assisting in the detection of diagnostic biomarkers. Emerging indicators for gut microbiota health and function include indole and its derivatives, including tryptamine and indoxyl sulfate [73]. They carry out key functions in the gut, such as indole acting as an intracellular signalling molecule that enhances epithelial barrier function [74], promoting the growth of beneficial bacteria while inhibiting the growth of harmful bacteria [75]. High indoxyl sulfate levels are associated with reduced bacterial diversity and shifts in bacterial diversity and composition [76]. Due to their functional roles in maintaining a healthy gut, measuring systemic levels of indole and its related metabolites provides insight into the gut microbiota functional status [77]. Currently, indole levels are measured by mass spectrometry or high-performance liquid chromatography (HPLC) coupled with mass spectrometry [78,79,80]. However, its potential as a point-of-care test has been limited due to its costly nature, the extensive sample preparation required, and long analysis times [81]. To combat this, a nanotip array has been proposed. It offers a rapid, simple, and low-cost electrochemical sensing alternative with minimal sample preparation required and a potential for miniaturisation [77]. The nanotip array provides a porous surface area for the attachment of selective capture molecules. Modification with silver (Ag) nanoparticles takes advantage of plasmonic effects to amplify electrical signals, allowing the specific and sensitive quantification of indole, tryptamine, and indoxyl sulfate via cyclic voltammetry and differential pulse voltammetry. Given the nanotip’s ability to analyse serum and faecal extracts, it has become extremely attractive for point-of-care testing and bears potential for wide usage [77] (Figure 2A and Table 1).

**Table 1 ijms-26-06465-t001:** Summary of nanomedicine based diagnostic approaches for IBD and CRC.

Application	Disease	Nanotechnology Used	Mechanism	Advantages	Stage	Reference
Biomarker Detection	General dysbiosis	Ag nanoparticles nanotip array	Electrochemical sensing	Low cost, simple preparation	Preclinical	[77]
	CRC	Monolayer-capped AuNPs	Electrochemical sensing	Quick, simple, disturbance-resistant		[82]
Diagnostic Imaging	IBD	SPIONs and In_2_O_3_ particles	MRI	MRI enhancement, tracks disease activity		[83]
		Dex-CeNP	CT imaging	Targets disease, enhances CT contrast		[84]
		HCFA	Hypoxia-activatable fluorescence probes	Detects hypoxia degrees for precision therapy		[85]
	CRC	AuNPs	Detect MGF in CRC tissues	Identifies cancer; stronger, more stable emission		[86]
		SPIONs	MRI contrast agent	MRI enhancement		[87]
		Quantum dots	Contrast agent for fluorescence imaging	Size-modulated absorbance and emission, high photostability, longer excited state etc.		[88]
		PANAM dendrimers conjugated with various aSlex antibodies	Detects circulating tumour cells	High capture, sensitive, noninvasive prognostic tool		[89]

#### 5.1.2. Biomarkers in CRC

More specific to CRC is a nanosensor array developed using monolayer-capped 5 nm gold nanoparticles (AuNPs), used as a breath sensor. It bears the capability to distinguish between the breaths of healthy and cancerous patients and even to differentiate the types of cancers based on their specific volatile organic compounds (VOCs). This breath analysis method had an edge over breath analysis via gas chromatography linked to the mass spectrometry technique (GC-MS) due to its minimal preparation (pre-concentration or de-humidification) needed, quick and simple procedure, and insensitivity to external interrupting factors which could potentially alter the chemical composition of test subject’s breath, making it a technology that could have significant beneficial impact on cancer treatment [82] (Table 1).

### 5.2. Nanomedicine in Diagnostic Imaging

Nanotechnology has also proven its utility in radiography modalities through aiding in early diagnosis and improving the prognosis of specific diseases.

#### 5.2.1. Diagnostic Imaging of IBD

In IBD, the use of macrophages labelled with superparamagnetic iron oxide (SPIO) and indium (111) oxide (In_2_O_3_) nanoparticles enhanced magnetic resonance imaging (MRI) quality by visualising areas on the intestinal wall where the signal was originally lost. The percentage of normalised enhancement at MRI correlated well with disease activity, revealing the utility and reliability of the incorporation of SPIO nanoparticles (SPIONs) in MRI-based techniques as a better way to study and monitor IBD activity [83]. Another developing nanotechnology developed to target IBD is dextran coated cerium oxide nanoparticles (Dex-CeNP). Unlike the iodine and barium-based contrast agents used in hospitals today, Dex-CeNP can enhance CT contrast generation and accumulate in inflammation sites, allowing for the localisation of colitis sites in mouse models. They also carry the additional benefit of being able to protect cells against oxidative damage in vitro. Dex-CeNPs have shown potential to be transferred to human clinical use, given that 99.9% and 97.6% of the nanoparticles were cleared within 24 h in healthy and colitis mice, respectively [84]. Additionally, the more moderate hypoxic state of patients with IBD compared to other diseases has posed a challenge for traditional hypoxia-activatable fluorescence probes. This has prompted the development of hypoxia-activatable and cytoplasmic protein-powered fluorescence cascade amplifiers (HCFA), which were successful in distinguishing the varying degrees of cellular hypoxia sensitively—a significant advancement over conventional diagnostic tools that lack sensitivity in detecting subtle hypoxic variations, enabling greater precision in IBD diagnosis and management [85] (Figure 2B and Table 1).

#### 5.2.2. Diagnostic Imaging of CRC

Diagnosing CRC has also been a beneficiary of developments in nanotechnology. Near-infrared-emitting AuNPs have been found useful in investigating mechano-growth factor (MGF), an insulin-like growth factor (IGF)-1 that is overexpressed in colon cancer tissues. Moreover, unlike traditional fluorescent dyes, AuNPs show stronger emission intensity and photostability and are not cytotoxic like semiconductor quantum dots, offering a safer and more durable alternative for imaging [86]. Nanoparticles are also promising contrast agents for MRI, with SPIONs coated with a polymer demonstrating such potential [87]. Quantum dots have also been a growing platform for cancer research and can be used as an alternative to organic dyes. Some of their favourable properties include size-modulated absorbance and emission, high photostability, a longer lifetime of an excited state, and more [88]. These help to overcome the limitations of today’s conventional dyes, which cannot be precisely tuned and undergo quick degradation. Nanotechnology can also be used to detect circulating tumour cells, which are tumour cells shed from the primary tumour that drive cancer relapse. The concentration of such cells is normally very low, making them difficult to detect and use to determine patient prognosis and the chances of relapse. Poly(amidoamine) (PANAM) dendrimers were conjugated with various purified anti-Slex (aSlex) antibodies and were able to detect HT29 colon cancer cells with a maximum capture efficiency of 77.88% obtained within 1 h of exposure. Notably, the nanoparticles could even detect HT29 cells when diluted with other cells commonly found in blood, demonstrating a significant improvement in detection sensitivity and potential for earlier intervention [89] (Figure 2C and Table 1).

## 6. Therapeutic Nanomedicine

### 6.1. Nanomedicine in Drug Delivery

Nanoscale drug delivery systems (DDS) have been gaining much traction recently due to their ability to carry out targeted drug delivery, maximising efficacy while minimising undesired side effects. This can be attributed to the nanoscale materials allowing for the customisation of specific DDS properties, including drug release characteristics, dissolution, solubility, bioavailability, and immunogenicity [90]. The utility of nanomedicine in drug delivery in the realm of treating gut dysbiosis and its associated diseases is discussed below.

#### 6.1.1. Drug Delivery for IBD

Patients with IBD have been found to have significantly heightened concentrations of reactive oxygen species (ROS) compared to normal mucosa, providing a potential target for DDS [91]. Based on this concept, a new nanoparticle prodrug named Bud-ATK-Tem (B-ATK-T) has been designed, which links budesonide together with tempol via aromatised thioketal. These thioketal bonds break down in the presence of excessive ROS at inflamed areas, allowing drug release from 55–100% based on the level of ROS present. In a mouse model with dextran sulfate sodium-induced colitis to model IBD, the B-ATK-T nanoparticle also reduced colon inflammation and weight loss and improved the disease activity index while not exhibiting any toxicity to major organs [92] (Figure 3 and Table 2). This targeted ROS-responsive strategy offers an edge over systemic corticosteroids by minimising off-target exposure and reducing systemic toxicities.

To overcome the problems of the current first line therapy 5-ASA, including a high pill burden (2–4x dosing daily) and systemic adverse effects [93], a chitosan (Csn) bound ginger nanocarrier has been developed to enhance site-targeted drug delivery to inflamed mucosal linings in IBD. Csn is formed via the deacetylation of chitin, and it has been recognised for its non-toxicity, biocompatibility, and biodegradability. The drug carrier complex was able to entrap an estimated 50% of the drug. At pH 1.2, the amine groups of Csn chain are protonated, causing the Csn chain to relax due to mutual electrostatic repulsion. This results in the swelling of nanocarriers, allowing large amounts of water molecules to enter the nanoparticle network, causing drug dissolution. Such is the mechanism of pH-dependent drug release. Nanotechnology thereby addresses the limitations related to drug retention and patient compliance in conventional 5-ASA therapy [94] (Figure 3 and Table 2).

The delivery of corticosteroids via nanoparticle DDS has also helped to reduce its systemic effects of immunosuppression, potentially enabling safer long-term use in IBD treatment. For instance, the use of pH-sensitive Eudragit L100–55 polymer microparticles loaded with prednisolone ensures that it is only released within the colon, and not in the upper GI tract, where the pH is acidic [95]. Tannic acid, a degradable mucoadhesive polyphenol, together with polymers, was formulated to form polyphenols and polymers self-assembled nanoparticles (PPNPs) made to carry out targeted drug delivery to the colon. Esterase, whose levels are elevated in colon inflammation, can hydrolyse the polyphenols and enable drug release. Comparing the levels of dexamethasone released in the colon without and with esterase, its release increased from 30% to 62%. This demonstrates that the approach of incorporating polyphenol in corticosteroid therapy not only improves targeted drug delivery, but also offers enzyme-responsive release for greater therapeutic control [96] (Figure 3 and Table 2).

#### 6.1.2. Drug Delivery for CRC

Nano-targeted DDS have also gained popularity among research in cancer pharmacology. Nanoparticles can accumulate in tumour cells due to the enhanced permeability and retention (EPR) effect, where the leaky tumour vasculature results in increased permeability to nanoparticles [97]. Cancer cells also rely on glycolysis instead of the Krebs cycle to produce ATP, known as the Warburg effect, causing the tumour environment to be acidic due to the lactic acid formed as the tumour cells undergo anaerobic respiration [97]. These specific tumour environments allow for drugs to be released at the tumour site, minimising systemic toxicity and improving therapeutic selectivity, an edge over current conventional therapy [98]. Bufadienolides nanocrystals decorated by Csn quaternary ammonium salt could protect bufadienolides from damage in the acidic environment of the stomach, increasing drug delivery to the tumour sites in the colon and increasing its therapeutic effect [99].

Nanoparticles can also be modified and functionalised with specific ligands that enhance their ability to target and bind to CRC cells, thereby improving their uptake by these cells. An epidermal growth factor (EGF) modified poly (lactic-co-glycolic acid) (PLGA) nanoparticle was loaded with 5-fluorouracil (5-FU) and perfluorocarbon (PFC) for a targeted treatment for CRC [100]. The presence of EGF allowed the nanoparticles to interact strongly with cancer cells that expressed EGFR, as the cancer cells could take up the cells via EGF receptor-mediated endocytosis. Nanoparticles with EGF had a more pronounced tumour suppression of the SW260 colon cancer cell line and also induced apoptosis in more colon cancer cells [100]. The epithelial cell adhesion molecule (EpCAM) is another target for drug delivery, as it is a dominant surface antigen in human colon carcinoma [101]. Polyethylene glycol (PEG) dendrimers with EpCAM aptamer were able to deliver celastrol, an anti-cancer agent, to induce apoptosis in SW260 colon cancer cells in vitro and in mice [101]. This targeted DDS was able to reduce local and systemic toxicity in xenograft and mice and zebrafish models, demonstrating a safer and more effective treatment approach [101] (Figure 4 and Table 2).

Beyond targeting the cancer cells themselves, scientists have begun looking into treating cancer stem cells (CSCs). CSCs are a subset of cancer cells with increased renewal capacity and the ability to recapitulate the heterogeneity found in primary tumours, playing an essential role in tumour growth and metastasis [102]. Yet conventional therapies can fail to eliminate these CSCs, giving tumours the chance to relapse and develop drug resistance. In view of improving the delivery of targeted chemotherapy to these cells, there has been a development of a biocompatible tumour-cell-exocytosed exosome-biomimetic porous silicon nanoparticle (PSiNPs). These nanoparticles, when loaded with doxorubicin (DOX), have exhibited enhanced tumour accumulation, tumour penetration, and cross-reactive cellular uptake by both bulk cancer cells and CSCs. Intracellular adhesion molecule 1, a member of the immunoglobulin supergene family, was found to be involved in the cross-reactive cellular uptake of the PSiNPs by cancer cells. Notably, killing activity was demonstrated across several tumour models, including subcutaneous transplantation and orthotopic and advanced metastatic tumour models, thus addressing a critical gap in therapy by targeting CSCs [103] (Figure 4 and Table 2).

### 6.2. Nanomedicine in Targeted Therapies

#### 6.2.1. Hyperthermia Treatment for CRC

Hyperthermia treatment at 43 °C for 60 min can inactivate tumour cells [104]. The greatest therapeutic effect is observed when hyperthermia is combined with the administration of local chemotherapeutic agents, maximising cytotoxicity to the tumour area [105]. Specifically, hyperthermia increases 5-FU efficacy in disrupting DNA synthesis and repair mechanisms [106]. Csn-coated magnetic nanoparticles (MNPs) with 5-FU were injected into human colon cancer xenografts in mice before the tumours were exposed to an alternating magnetic field. The thermo-chemotherapeutic treatment resulted in distinct tumour regression, much more evident than seen when magnetic hypothermia or 5-FU monotherapy were used [104]. 5-FU loaded onto PLGA encapsulating iron oxide nanoparticles has also been found to have increased cytotoxic activity on human colon cancer HT-29 cell lines when exposed to hyperthermia. The nanoparticles combined with hyperthermia treatment were found to have a cytotoxicity of 97.3%, compared to 75.5% when 5-FU was given alone. This synergistic effect enables the administration of lower drug doses, thus potentially minimising the adverse effects caused by conventional chemotherapy [107] (Figure 4 and Table 2).

#### 6.2.2. Magnetic Drug Targeting for CRC

MNPs have several advantages over nonmagnetic nanoparticles, as drug release can be controlled via specific magnetic field stimulation, thus improving drug stability and bioavailability. They can also be localised and targeted to specific tissues, thus decreasing systemic exposure and minimising off-target side effects associated with current day chemotherapy [108]. SPIONs can aggregate to tumour sites upon exposure to a magnetic field, reducing uncontrolled aggregation in other tissue or in the bloodstream. Docetaxel, a drug that is not systemically used due to its hydrophobic nature, was encapsulated into the oil core of polymeric SPIONs. The nanoparticles were found to have an efficient cell-killing effect on the CT26 cell line in an in vitro cytotoxicity study. Only the magnetically targeted groups were found to have significantly smaller tumour volume compared to that of control treatments [109]. SPIONs were also aggregated with mAb198.3, an antibody that can stain and recognise CRC samples from surgically resected samples. The SPIONs linked to mAb198.3 significantly reduced tumour growth compared to the control and the nanoparticles without mAb198.3. A similar tumour size was found for the SPIONs linked to mAb198.3 compared to just using free mAb198.3, despite a 300-fold lower concentration used when the antibodies were linked to SPIONs. Erythrocytes were engineered to contain the SPIONs, and tumour reduction was noted earlier than with SPIONs alone, indicating enhanced delivery efficiency and therapeutic result. This is because erythrocytes can protect the loaded drug and reduce the possibility of a severe immunologic reaction. [110] (Figure 4 and Table 2).

EGFR-targeted monoclonal antibodies like Cetuximab have improved survival rates in patients, but their effectiveness has been limited by the KRAS activating mutations that confer resistance in approximately 25% of advanced CRC cases [111]. MNPs can induce oxidative stress by depleting glutathione and inducing ROS production [112]. This ROS-dependent apoptosis is found to enhance existing CRC treatment given drug-resistant late-stage cases. Combining Cetuximab and MNPs showed that approximately 31% of SW-480 CRC tumour cells exhibited signs of apoptosis, higher than the 10% observed when Cetuximab was used alone. This synergistic action led to a significant increase in ROS production, suggesting that intracellular ROS production did lead to the cytotoxic effect observed and holds promise of combining Cetuximab and MNPs to treat cancer [111] (Figure 4 and Table 2).

#### 6.2.3. Photothermal Therapy (PTT) for CRC

Photothermal therapy (PTT) is an emerging and promising strategy in cancer treatment. It involves the use of photothermal agents (PTAs) which convert light energy—typically in the near-infrared (NIR) region—into heat energy, inducing localised thermal injury to tumour tissues and ultimately leading to cancer cell death. An advantage of PTT over conventional cancer therapies is its highly effective and non-invasive nature [113], but a challenge still lies in the systemic distribution of PTAs in the body, which leads to non-precision exposure and damage to normal tissues surrounding the tumours [114]. Nanotechnology has significantly advanced in the field of PTT. Apart from facilitating cancer diagnosis and treatment monitoring through improved imaging capabilities, nanoparticles have also demonstrated strong absorption in the NIR region alongside accurate targeting of cancer cells for improved specificity, making them highly suitable candidates for PTT applications [115]. Commonly used nanoparticles include metal (gold, iron oxide, Ag etc.) nanoparticles, polymer nanoparticles, and carbon nanoparticles.

AuNPs are able to convert light energy in the NIR region to heat energy effectively, making them suitable for PTT as NIR is able to penetrate tissue optimally [116]. AuNPs have good biocompatibility and can be functionalised with targeting ligands. PEG-ylated nanoshells with gold were able to significantly reduce rapid tumour growth when tumours were irradiated with NIR light in CT26 CRC tumour xenografts in murine models [117]. AuNPs can also be functionalised with an A33 antibody that binds to the surface antigen A33, which is overexpressed in colon cancer, providing a more targeted PTT approach [118]. Gold nanorods were also able to prolong the lifespan of mice bearing CT26 CRC xenografts, and the gold nanorods were found to accumulate mainly in tumours, with no confirmed toxicity to other organs [119] (Figure 4 and Table 2).

Carbon nanotubes are also able to convert NIR light energy to heat, making them a suitable candidate to be used for PTT [116]. Carbon nanotubes can be functionalised with nanocomposite polyhedral oligomeric silsesquioxane poly (carbonate-urea) urethane (POSS-PCU), and can reduce the viability of HT29 CRC cells by 95% upon irradiation with NIR light [120]. Carbon nanotubes can also be conjugated with folic acid onto the surface, as the folic acid receptor is overexpressed in CRC. Cell viability was also found to be greatly reduced in HTC116 and RKO CRC cells after irradiation with NIR light [121]. The combination of PTT with immunotherapy has been explored with the development of GNC-Gal@CMaP nanocomposites. They consist of hollow gold nanocages (GNCs) loaded with galunisertib, a transforming growth factor (TGF)-β inhibitor, and surface-functionalised with anti-programmed cell death ligand-1 (PD-L1) antibodies. The study showed that GNC-Gal@CMaP selectively accumulated in cancer cells pretreated with anti-PDL1, was able to eliminate the primary tumour mass under NIR light irradiation, and inhibited distant metastases, demonstrating the abscopal effect [122] (Figure 4 and Table 2).

A multifunctional endoscope-based interventional system consisting of both bioelectronics and nanoparticles has also been developed. These nanoparticles are engineered to carry both phototherapeutic and chemotherapeutic agents, which can be precisely delivered and locally activated via light stimulation. This system offers optical fluorescence-based mapping, radio frequency-based ablation, and site-specific photo/chemotherapy, forming the basis of a minimally invasive treatment strategy for CRC [123] (Table 2).

### 6.3. Nanotechnology and the Gut Microbiota

Nanoformulations of probiotics, prebiotics, and synbiotics are emerging as powerful tools for optimising gut health treatments [124].

#### 6.3.1. Nano-Prebiotics

Prebiotics are non-digestible food ingredients that selectively stimulate activity in beneficial bacteria in the colon. Prebiotics can be used as a vehicle for the improved delivery of nanomedicine. Pectin (Pcn) is a biocompatible polysaccharide that has a positive reputation for biodegradability, biocompatibility, and non-toxicity [125]. Csn is also a biopolymer that has received attention due to its biocompatibility and antimicrobial [126], mucoadhesive, and absorption-enhancing properties [127]. Together, they were both used as a prebiotic shell around a PLGA core loaded with sulfasalazine. This design exhibited pH-responsive characteristics for protecting anti-inflammatory drugs in the upper GIT, and also demonstrated a specific colonic delivery property triggered by pectinase, which degraded the Pcn/Csn shell, whose enzymatic product revealed notable prebiotic properties [128]. High molecular weight insulin nanoparticles were also shown to improve drug delivery, with no toxicity detected in peripheral blood mononuclear cells at concentrations < 200 μg/mL [129] (Table 2). Other nano-prebiotics, such as phthalyl dextran nanoparticles, have also shown antimicrobial effects [130] (Figure 5). These nano-prebiotic systems thus overcome the limitations of conventional prebiotic supplementation through offering targeted drug delivery, improved drug stability, and microbiome-modulating properties.

#### 6.3.2. Nano-Probiotics

Probiotics are live microorganisms (eg. bacteria, yeast) that offer health benefits to the gut microbiome when consumed. Nanotechnology has contributed to the development of strategies to preserve probiotic viability and functionality in food systems. A recent study developed modified poly(L-glutamic acid) (PG) films incorporating varying levels of poly(L-lysine) (PL), resulting in a dual-function active packaging material. It serves as a microbioreactor, which enabled continued gamma-aminobutyric acid (GABA) production by the probiotic during the shelf life of the packaged food. This can be attributed to its stability, low toxicity, and biodegradability. It also demonstrates antimicrobial activity, preventing the growth of foodborne pathogenic bacteria [131] (Figure 5 and Table 2).

Nano-probiotics have demonstrated utility in their anti-cancer properties. Synthesised Ag/*Lactobacillus rhamnosus* GG nanoparticles (Ag-LNPs) were assessed against CRC cell lines and significantly decreased the viability of these cell lines from 7.8–1000 μg/mL, possibly due to the activation of ROS, which led to cell damage and death [132]. The probiotic bacteria *Lactobacillus casei* was also used to synthesise copper oxide nanoparticles (CuONPs) that exerted cytotoxic effects by suppressing growth, increasing oxidative stress, and inducing apoptosis on cancer cells, alongside antimicrobial effects on gram positive and negative bacteria [133] (Table 2). Other nanotechnologies that have shown anti-cancer properties include selenium nanoparticles synthesised by *Lactobacillus casei* [134,135], dead nano-sized *Lactobacillus plantarum* (LP) [136], and AuNPs synthesised by novel probiotic *Lactobacillus kimchicus* [137] (Figure 5). These technologies highlight how nano-probiotics can go beyond conventional supplementation to have bioactive and anti-cancer effects in the gut.

#### 6.3.3. Nano-Synbiotics

Synbiotics are combinations of probiotics and prebiotics that work synergistically to improve host health. Nano-prebiotics [eg. phthalyl pullulan nanoparticles (PPNs)] have been developed to enhance the antimicrobial action of probiotics (LP) and were tested together as synbiotics for their therapeutic effect on dysbiosis-induced models. Results demonstrated that antimicrobial activity of the PPN-internalised LP was significantly higher than that of untreated LP or pullulan alone. Additionally, groups that underwent synbiotic treatment showed the greatest body weight gain and longest colon lengths [138], preclinical markers of therapeutic efficacy in IBD animal models. Body weight gain reflects reduced disease activity and improved systemic health, while the preservation of colon length indicates the attenuation of inflammation and protection of intestinal tissue. Administering synbiotics with PPNs limited the influx of endotoxins into blood, strengthening the gut barrier. This reduced coliform bacteria while boosting lactic acid bacteria, restoring microbiome diversity to a more balanced state [138] (Figure 5 and Table 2).

Nano emulsion systems have also been developed to enhance bioavailability and maintain the viability of probiotics. One example is the formulation of a synbiotic nanoemulsion incorporating whey protein concentrate, inulin, Gum Arabic, and *Enterococcus faecium* in the aqueous phase, with coconut oil as the oil phase. The study showed that the droplet size increased up to 500 nm after 60 days of storage, while the viability of the probiotic organism remained promising [139] (Figure 5 and Table 2).

Encapsulation technologies have also been explored to enhance the tolerance and stability of probiotic bacteria. A study developed an oral synbiotic supplement using alginate beads incorporated with varying concentrations of inulin to protect three probiotic strains: *Pediococcus acidilactici*, *Lactobacillus reuteri*, and *Lactobacillus salivarius*. Notably, encapsulation did not affect the antimicrobial or probiotic properties of the strains. Inulin-enhanced beads improved bacterial survival in acidic conditions, while beads containing 5% inulin offered the greatest protection against bile salts [140]. These advances in nano-synbiotic delivery have shown an improvement in functional outcomes compared to traditional formulations, offering novel treatment for gut dysbiosis (Figure 5 and Table 2).

#### 6.3.4. Nanoparticles with Antioxidant and Anti-Inflammatory Effects (IBD)

Tantalum (Ta), a medical metal element known for its superb physicochemical properties, has been used for the treatment of many diseases, but yet to be explored deeply in IBD. Modification with chondroitin sulfate has resulted in the formation of TACS (Ta2C modified with chondroitin sulfate), evaluated to be a highly targeted therapy nanomedicine for IBD. Its main mechanisms of action are mitochondrial protection, oxidative stress elimination, and the inhibition of macrophage M1 polarisation, among others [141] (Figure 6 and Table 2). This multitargeted mechanism and high delivery specificity give TACS a therapeutic edge over traditional anti-inflammatory agents, which often act broadly and risk higher systemic adverse effects.

Another therapeutic strategy being explored is nanomedicine that treats IBD by targeting multiple pro-inflammatory factors that contribute to its pathogenesis. Specifically, researchers reasoned that the neutralisation of cell-free DNA (cfDNA) could modulate the imbalanced immune response in IBD [142]. This is because cfDNA released by damaged cells activated toll-like receptor (TLR)-9 mediated pro-inflammatory signalling in immune cells and contributes to the extent and timespan of inflammation [143]. A nanoparticulate, polyethylenimine-mesoporous organosilica nanoparticles (MON-PEI), was developed via conjugating DNA binding polyethylenimine to antioxidative diselenide-bridged mesoporous organosilica nanoparticles (MON), which bears cfDNA-scavenging, antioxidative, and anti-inflammatory properties. It also effectively decreased cfDNA-induced TLR9-MyD88-NF-κB signalling and pro-inflammatory macrophage activation in treating inflammation. Importantly, it exhibited a lower dose frequency with a better safety profile than that of mesalazine, due to its preferential accumulation in diseased areas [142] (Figure 6 and Table 2). This illustrates how nanomedicine can enhance therapeutic efficacy while reducing systemic burden.

Another approach involves capitalising on physiological anti-inflammatory molecules such as the gut hormone glucagon-like peptide-1 (GLP-1). In order to overcome its limitation of a short half-life due to degradation by enzymes such as dipeptidyl peptidase IV (DPP-4), GLP-1 in sterically stabilised phospholipid micelles (GLP-1-SSM) was developed. GLP-1-SSM treatment was able to lessen colonic inflammation and associated diarrhoea in mouse colitis models via decreasing the expression of IL-1β, a pro-inflammatory cytokine, increasing goblet cells, and preserving the architecture of the intestinal epithelium [144] (Figure 6 and Table 2). Encapsulation in nanocarriers not only preserved the bioactivity of GLP-1 but also enabled its sustained action in the gut, a key improvement over the rapid degradation seen in today’s free hormone administration.

Mimicking the body’s natural enzymes can also be a way to combat ROS. In this system, the free radical scavenger tempol, which acts as a superoxide dismutase (SOD) mimic, is loaded into a β-cyclodextrin-derived material (OxbCD) that also mimics catalase activity. These components were then assembled into stable nanoparticles with 1,2-distearoyl-sn-glycero-3-phosphoethanolamine (DSPE)-PEG. Upon entering inflamed areas with high levels of ROS, the nanoparticles released tempol, reducing both inflammatory cytokines and the disease index. It is significant to note that this nanoparticle treatment performed more effectively than free tempol and tempol-loaded PLGA nanoparticles [92] (Figure 6 and Table 2). This nanocarrier demonstrated its superiority to conventional therapy by achieving capabilities not achievable with free drug administration, including protecting therapeutic agents, bioavailability enhancement, and stimuli-responsive drug release.

#### 6.3.5. Nanomedicine for Microbiome Modulation

In IBD

Traditional approaches to treating IBD focus mainly on suppressing the immune response via immunosuppressant drugs. However, such a strategy causes systemic side effects and is unable to treat the disrupted intestinal barriers or restore the underlying gut dysbiosis. The development of hyaluronic acid-bilirubin nanomedicine (HABN) aims to improve this by accumulating in the inflamed colon and restoring the epithelium barriers in murine models of acute colitis [145]. HABN was also able to regulate gut microbiomes, increasing their diversity and richness. This is due to hyaluronic acid’s ability to regulate macrophages and induce antimicrobial peptides and CD4 T regulatory cells. Bilirubin has strong ROS scavenging, antioxidant, and cytoprotective properties that allow HABN to aggregate to colon inflammation. Mice undergoing HABN treatment were able to achieve full body weight recovery, maintain colon length, reduce colon damage, and increase beneficial bacteria in IBD such as *Akkermansia muciniphila*, *Clostridium,* and *Lactobacillus* [145] (Figure 6 and Table 2). These multi-target effects show how nanomedicine offers a more comprehensive treatment approach compared to conventional monotherapies.

Ag nanoparticles conjugated with a selective wall-binding domain targeting *Fusobacteriaceae* were found to eliminate these bacteria—known to be overabundant in IBD patients—and thereby reduce intestinal inflammation [146]. By selectively eliminating pathogenic microbes, these nanoparticles offer greater precision over conventional broad-spectrum antibiotics which risk disrupting the overall microbiome. Zinc oxide nanoparticles have also been found to reduce the population of *Lactobacillus* and alter their production of SCFA, a key process that is implicated in the pathogenesis of IBD [146] (Figure 6 and Table 2).

In CRC

Traditional methods to modulate the gut microbiome have shown promise in enhancing cancer treatment but damage the commensal gut microbiome [147]. The ability to modify the surface of nanoparticles allows them to target specific bacteria, resulting in a controlled killing of cancer-causing microbes with minimal off-target effects on beneficial gut flora [147]. A bacteriophage was found to eliminate a cancer-causing bacterium, *F. nucleatum*. The bacteriophage was functionalised to facilitate click-chemistry based linkage to nanoparticles. It was able to reduce *F. nucleatum* load with minimal inhibition to other bacteria strains, preserving strains such as *Clostridium butycirum*, that are useful in suppressing the growth of colon cancer [147] (Figure 7 and Table 2). This highlights how nanotechnology enables selective microbial targeting—a significant advancement over traditional antimicrobials or chemotherapy that often lack such specificity.

**Table 2 ijms-26-06465-t002:** Summary of nanomedicine based therapeutic approaches for IBD and CRC.

Application	Disease	Nanotechnology Used	Mechanism	Advantages	Stage	Reference
Drug delivery	IBD	B-ATK-T nanoparticle prodrug linked with budesonide	Thioketal bonds between budesonide and temporal degrade in ROS-rich areas	Targeted high dose delivery, reducing systemic side effects	Preclinical	[92]
		Csn bound ginger nanocarrier linked with 5-ASA	pH-sensitive drug carrier complex that facilitates colon-specific drug release	Site-specific delivery, lowering pill burden, and systemic exposure		[94]
		Eudragit polymer microparticles containing prednisolone	pH-sensitive drug carrier complex that facilitates colon-specific drug release	Localised immunosuppression, minimising systemic effects		[95]
		PPNP loaded with dexamethasone	Esterase-responsive systems trigger phenol hydrolysis and drug activation	Localised immunosuppression, minimising systemic effects		[96]
	CRC	EGF modified PLGA nanoparticles loaded with 5-FU and PFC	EGFR-targeting nanoparticles enable direct interaction with CRC cells	Enhanced tumour suppression and apoptosis induction		[100]
		PEG dendrimer nanoparticles with EpCAM aptamer loaded with Celastrol	EpCAM aptamer functionalised nanoparticles selectively bind to cancer cells	Improved drug precision with reduced local and systemic toxicity		[101]
		PSiNPs loaded with DOX	Nanocarriers enhance tumour accumulation and penetration in CSCs	Improved chemotherapy efficacy		[103]
Hyperthermia treatment	CRC	Csn-coated MNPs with 5-FU	Hyperthermia with chemotherapy enhances tumour regression	Improved chemotherapy efficacy		[104]
		5-FU loaded onto PLGA encapsulating iron oxide nanoparticles	Increased cytotoxic activity against colon cancer cells	Lower effective doses, reducing systemic toxicity		[107]
Magnetic drug targeting	CRC	Docetaxel encapsulated with oil core polymeric SPIONs	Magnetic field induced nanoparticle aggregation	Efficient cytotoxicity with precise delivery and minimal systemic side effects		[109]
		SPIONs aggregated with mAb198.3	mAb198.3 stains CRC cells	Significant tumour growth reduction		[110]
		MNPs combined with Cetuximab	MNPs induce oxidative stress, overcome Cetuximab resistance	Potential to overcome Cetuximab-resistant CRC		[111,112]
PTT	CRC	AuNPs functionalised with A33 antibody	A33-antibody functionalised PTAs absorb NIR light, targeting CRC tumour with effective accumulation and cancer cell death	Targeted PTT with no other organ toxicity		[117,118,119]
		Carbon nanotubes functionalised with nanocomposite POSS-PCU	Functionalised carbon nanotubes aggregate in CRC tumours, reducing cancer cell viability	Enhanced targeting of PTT through antibody functionalisation		[120,121]
		GNC-Gal@CMaP nanocomposites loaded with galunisertib, surface-functionalised with anti-PD-L1 antibodies	Tumour-selective nanoparticle accumulation for enhanced PTT	Antibody-functionalised PTT eliminates primary tumours while inhibiting metastases		[122]
		Multifunctional endoscope-based interventional system	Fluorescence mapping, radio frequency-based ablation, targeted photo/chemotherapy	Novel minimally invasive CRC treatment		[123]
Nano-prebiotics	IBD	Pcn and Csn prebiotic shell surrounding PLGA core loaded with sulfasalazine	pH-responsive prebiotic shell for drug protection and delivery	Improved drug concentration at target sites		[128]
		High molecular weight insulin nanoparticles	Improved drug delivery	No peripheral toxicity		[129]
Nano-probiotics	General dysbiosis	Modified PG films incorporating varying levels of PL	Active packaging sustaining probiotic GABA production with antimicrobial effects	Preserves probiotic viability and function throughout shelf life		[131]
	CRC	Ag-LNPs	ROS-induced CRC cell death	Reduced CRC cell viability		[132]
		CuONPs synthesised with *Lactobacillus casei*	Anti-cancer, pro-oxidative, apoptotic, antimicrobial	CRC cytotoxicity		[133]
Nanosynbiotics	IBD	PPNs and LP	Improved gut barrier and microbiome balance	Antimicrobial and anti-dysbiosis effects		[138]
		Nanoemulsion incorporating whey protein concentrate, inulin, Gum Arabic, and Enterococcus faecium, coconut oil	Improved probiotic stability and absorption	Enhanced probiotic efficacy		[139]
		Alginate beads incorporated with inulin to protect probiotic strains *Pediococcus acidilactici, Lactobacillus reuteri,* and *Lactobacillus salivarius*	Enhanced gut-targeted probiotic efficacy	Enhanced gut probiotic action		[140]
Antioxidant and anti-inflammatory nanoparticles	IBD	TACS	Anti-inflammatory and mitochondrial protection	Reduced IBD inflammation		[141]
		MON-PEI	Multitargeted anti-inflammatory properties with improved safety	Lower dosing and improved safety compared to mesalazine		[144]
		GLP-1-SSM	Preserves intestinal architecture and reduces colonic inflammation	Alleviates colonic inflammation and diarrhoea		[145]
		DSPE-PEG assembled with tempol and OxbCD	Targeted ROS-triggered anti-inflammatory action	Superior anti-inflammatory effect compared to free tempol		[92]
Microbiome Modulation	IBD	HABN	Targets gut inflammation, barrier repair, and dysbiosis	Restores gut microbiome and epithelial integrity		[145]
		Ag nanoparticles targeting *Fusobacteriaceae*	Precision targeting of pathogenic *Fusobacteriaceae*	Targets IBD-related dysbiosis		[146]
		Zinc oxide nanoparticles	Modulates SCFA levels and microbiome implicated in IBD	Modulates IBD pathogenesis		[146]
	CRC	Bacteriophage targeting *F. nucleatum* functionalised to nanoparticles	Selective elimination of oncogenic bacteria	Selective killing of oncogenic microbes		[147]

## 7. Challenges and Future Directions

While nanotechnology shows great promise in overcoming current limitations, it is not without challenges. One of the major challenges nanotechnology faces is its safety in humans due to its cytotoxic nature. However, it has been found that Ag nanoparticles have a safer profile for normal cells compared to cisplatin, an FDA approved compound for chemotherapy. This is shown by their difference in inhibitory concentration 50% (IC_50_) values for human embryonic kidney (HEK)-293 cells, a normal cell line. The IC_50_ value was 48.11 ppm and 2.5145 ppm on HEK-293 cells for Ag nanoparticles and cisplatin, respectively, demonstrating the enhanced safety profile of Ag nanoparticles [148]. The safety of nanoparticles has also been demonstrated in humans, using SPIO as an MRI contrast to detect hepatocellular carcinoma. The Phase II clinical trial found that SPIO injections in 52 subjects did not result in any severe adverse effects after 5 days of monitoring, showcasing the potential for such nanoparticles to be used in everyday practice [149]. There have been concerns about the long-term toxicity of nanoparticles, as they can accumulate in organs such as the liver and spleen [150]. AuroShell^®^ (Nanospectra, Houston, TX, USA) is a silica-gold shell nanotechnology that converts NIR light to heat. It has entered clinical trials to ablate low-risk prostate tumours, and was able to ablate 94% of treated tumours without significant adverse effects or organ damage [151]. Notably, patients showed no device related toxicities even after a year, demonstrating the potential feasibility and long-term safety of such nanotechnology-based treatments in humans [152]. Nanotherm, iron oxide nanoparticles used in hyperthermia treatment, has obtained European Union (EU) regulatory approval to treat glioblastoma and FDA approval to treat intermediate risk prostate cancer. These significant milestones reflect the recognised safety of such nanotechnologies to treat various types of cancers [153].

Currently, there is no nanotechnology treating IBD in clinical trials. With regards to CRC treatment, a phase I, II trial has been conducted. Nanoparticle albumin bound (nab)-sirolimus was administered with bevacizumab, folinic acid, 5-FU, and oxaliplatin to patients with metastatic colon cancer. 89% of patients experienced a reduction in tumour size, with 39% of patients experiencing a tumour shrinkage of more than 30%. 63% of patients experienced grade 3 to 4 treatment related adverse events, with 25% of patients experiencing neutropenia, and 17%, thrombocytopenia. The Phase II trial is currently still ongoing [154]. Another Phase II clinical study uses a polymeric nanoparticle comprised of a cyclodextrin-PEG copolymer conjugated to camptothecin to treat locally advanced rectal cancer. This nanoparticle is co-administered with capecitabine and radiotherapy. The treatment is well tolerated, supporting further evaluation for local control aligned with standard chemoradiotherapy regimens [154].

So far, only two clinical trials have used nanotechnology to treat CRC and none have used it for IBD. Hence, more clinical studies are still required to evaluate the safety and efficacy of these technologies in humans before they can be properly integrated into clinical practice. The success of the clinical trials described above serve as good groundwork for future research.

Some of the other challenges include difficulties with reproducible manufacturing and scale-up, inadequate characterisation tools, instability in vivo, safety concerns, and a limited understanding of disease variability and patient selection [155]. Moreover, regulatory agencies such as the United States FDA and European Medicines Agency (EMA) require extensive safety and efficacy evaluations for nanotechnology, yet there is a lack of a standardised protocol to determine the safety of nanomaterials [156]. Concerns have also been expressed over the potential long-term toxicity of nanoparticles due to their tendency to accumulate in organs such as the liver and spleen [156]. Furthermore, the GIT’s complex environment, characterised by fluctuating pH, enzymatic activity, and mucus layers, have also posed significant barriers for effective drug delivery [156], further complicating the transition of nanomedicine from lab research to clinical application. In essence, more extensive research, clinical trials, and standardised protocols are necessary to accelerate the integration of nanomedicine into routine clinical practice (Figure 8).

## 8. Conclusions

It is evident that gut dysbiosis plays a key role in the pathogenesis of IBD and CRC and should be treated to prevent implications in both the gut and other parts of the body. Nanomedicine has emerged as a transformative frontier, as it can overcome some of the limitations faced by existing treatments. Its unique chemical and physical properties allow for more sensitive and convenient biomarkers testing and also provide contrast that allows for greater detail on disease imaging. A significant advantage of nanotechnology is its ability to enhance treatment through targeted drug delivery, reducing off-target toxicities while maximising therapeutic efficacy. The versatility of nanomedicine is also shown through its many applications—in hyperthermia treatment, PTT, and magnetic targeting of specific tumours. Specific nanoparticles also bear anti-cancer and anti-inflammatory properties, while others have antimicrobial applications that can facilitate gut microbiome modulation. Hence, these nanoparticles have been found useful in targeting the underlying dysbiosis that underlies the pathogenesis of these diseases. While there are standing concerns regarding the safety and toxicity of these nanotechnologies, existing clinical trials have maintained a good safety record thus far. These studies serve as a foundation for other clinical trials to build on, as much more extensive testing on human beings is required for the clinical translation of nanomedicine into everyday treatment. Moreover, there is room for more research on how to incorporate real time information provided by nanotechnology diagnostics into clinical assessment to enhance treatment and prognosis. Clinical trials testing nanomedicines also need to incorporate them with pre-existing treatment methods to find the greatest synergistic effects to maximise patient outcomes. Alongside improved regulatory protocols and further technological advancements, we can expect nanomedicine to write new chapters of IBD and CRC treatment.

## Figures and Tables

**Figure 1 ijms-26-06465-f001:**
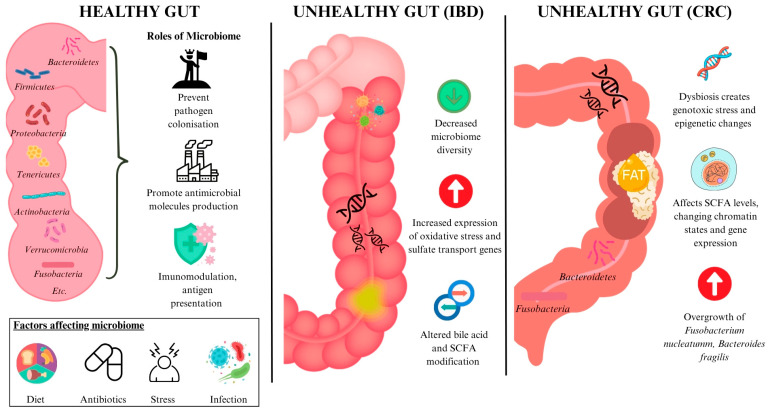
A comparison between a healthy gut and the diseased guts observed in IBD and CRC. Abbreviations: CRC: colorectal cancer; IBD: inflammatory bowel disease; SCFA: short chain fatty acid.

**Figure 2 ijms-26-06465-f002:**
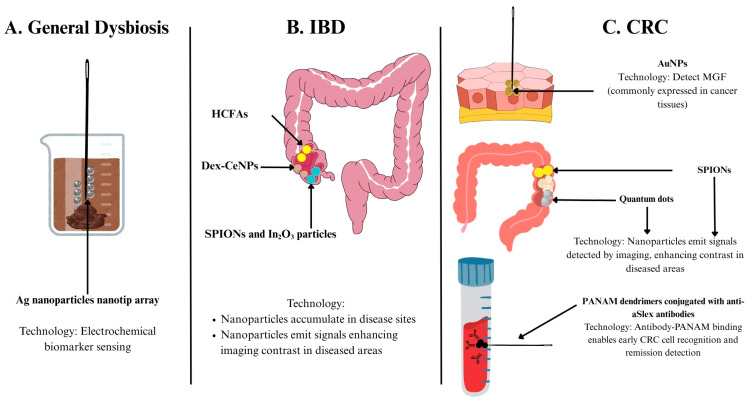
(**A**–**C**) Nanomedicine-enabled diagnostics for (**A**) general dysbiosis, (**B**) inflammatory bowel disease and (**C**) colorectal cancer. Abbreviations: Ag: silver; aSlex: anti-Slex; AuNP: gold nanoparticle; CRC: colorectal cancer; Dex-CeNP: dextran coated cerium oxide nanoparticle; HCFA: hypoxia-activatable and cytoplasmic protein-powered fluorescence cascade amplifier; IBD: inflammatory bowel disease; In2O3: indium (111) oxide; MGF: mechano-growth factor; PANAM: poly(amidoamine); SPION: superparamagnetic iron oxide nanoparticle.

**Figure 3 ijms-26-06465-f003:**
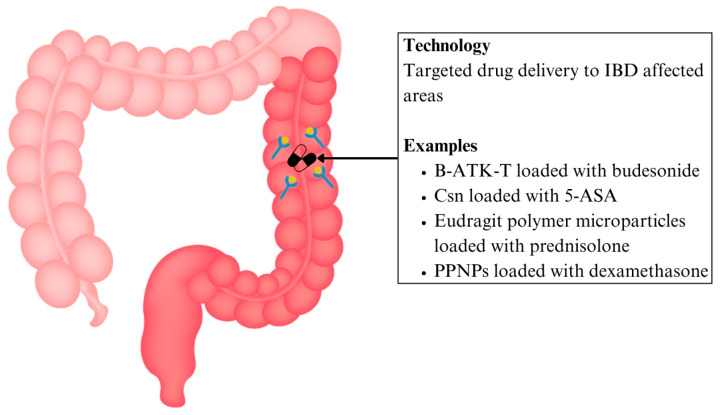
Targeted drug delivery systems are a form of nanotherapeutics for IBD. Abbreviations: 5-ASA: 5-aminosalicylic acid; B-ATK-T: Bud-ATK-Tem; Csn: chitosan; IBD: inflammatory bowel disease; PPNP: polyphenols and polymers self-assembled nanoparticle.

**Figure 4 ijms-26-06465-f004:**
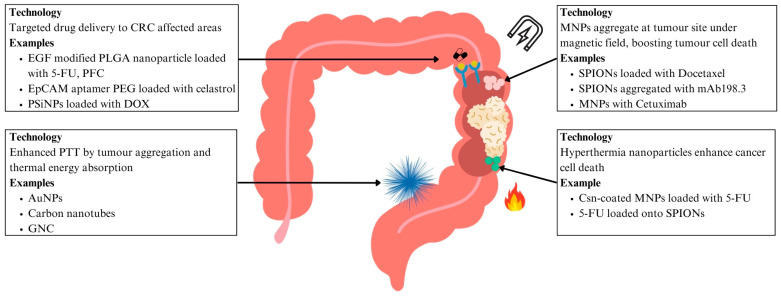
Nanotherapeutics for CRC. Abbreviations: 5-FU: fluorouracil; AuNP: gold nanoparticle; CRC: colorectal cancer; Csn: chitosan; DOX: doxorubicin; EGF: epidermal growth factor; EpCAM: epithelial cell adhesion molecule; GNC: gold nanocages; MNP: magnetic nanoparticle; PEG: polyethylene glycol; PFC: perfluorocarbon; PLGA: poly (lactic-co-glycolic acid); PSiNP: porous silicon nanoparticle; PTT: photothermal therapy; SPION: superparamagnetic iron oxide nanoparticle.

**Figure 5 ijms-26-06465-f005:**
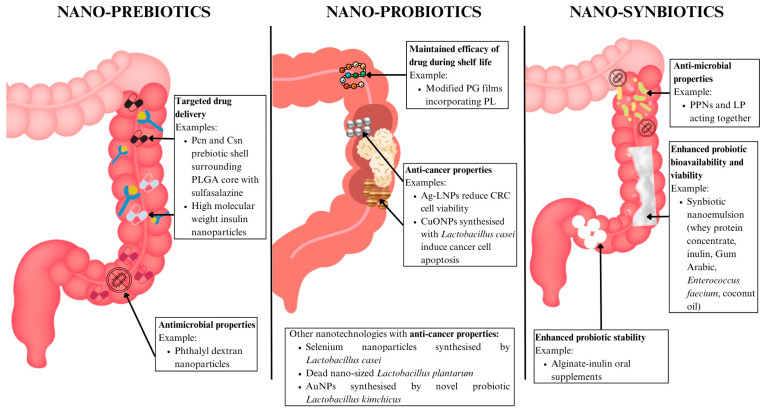
Nanotechnology-based microbiome modulators offer novel therapeutic avenues in IBD and CRC. Abbreviations: Ag-LNP: silver/Lactobacillus rhamnosus GG nanoparticle; AuNP: gold nanoparticle; CRC: colorectal cancer; Csn: chitosan; CuONP: copper oxide nanoparticle; LP: lactobacillus plantarum; Pcn: pectin; PG: poly(L-glutamic acid); PL: poly(L-lysine); PLGA: poly (lactic-co-glycolic acid); PPN: phthalyl pullulan nanoparticle.

**Figure 6 ijms-26-06465-f006:**
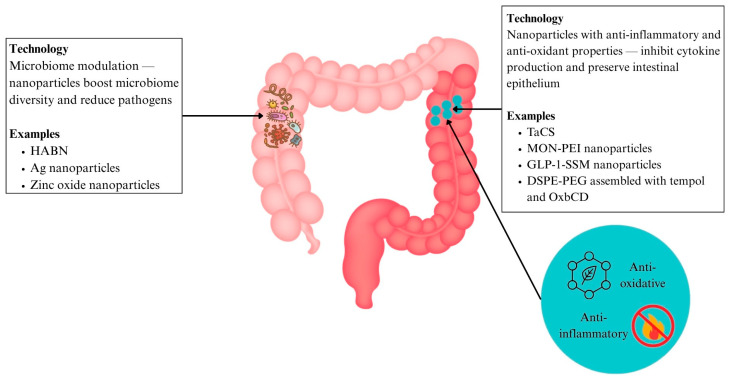
Nanotherapeutics in inflammatory bowel disease include nanoparticles with anti-inflammatory and antioxidant properties and microbiome modulation. Abbreviations: Ag: silver; DSPE: 1,2-distearoyl-sn-glycero-3-phosphoethanolamine; GLP-1-SSM: GLP-1 in sterically stabilised phospholipid micelles; HABN: hyaluronic acid-bilirubin nanomedicine; MON-PEI: polyethylenimine-mesoporous organosilica; OxbCD: β-cyclodextrin-derived material; PEG: polyethylene glycol; TACS: Ta2C modified with chondroitin sulfate.

**Figure 7 ijms-26-06465-f007:**
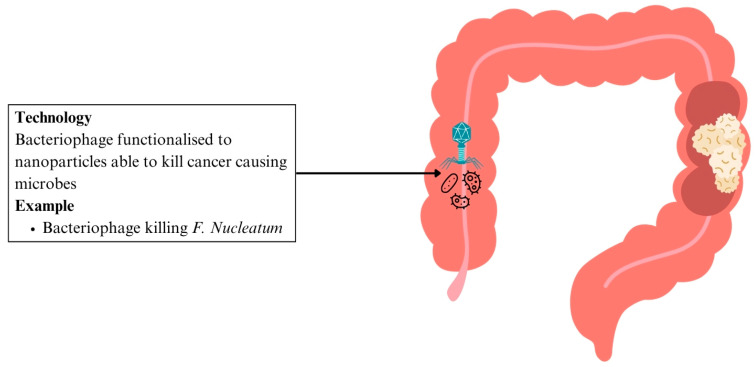
Nanotherapeutics in CRC include microbiome modulation. Abbreviation: CRC: colorectal cancer.

**Figure 8 ijms-26-06465-f008:**
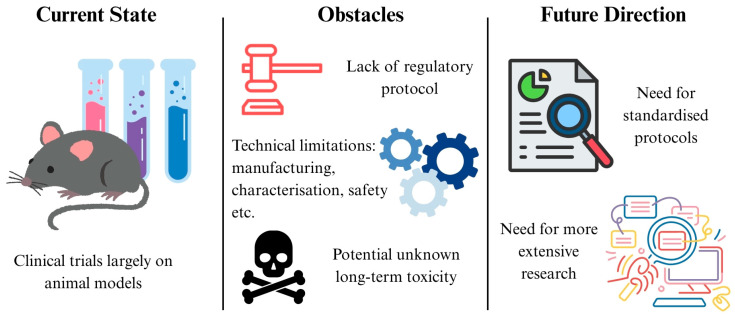
Roadmap for clinical translation of nanomedicine in IBD and CRC.

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
