# Peer review of "Nanomedicine Strategies in the Management of Inflammatory Bowel Disease and Colorectal Cancer"

_ijms, 2025, doi:10.3390/ijms26136465_

Round 1

Reviewer 1 Report

Comments and Suggestions for Authors

Comments to the Author

After reviewing the review, I believe that it is generally well-written and structured. Moreover, the review is technically sound. Only a few comments are necessary.

  1. The legends of the figures are very long and lack clarity, with some abbreviations left unexplained, which reduces the interpretability of the data. Please revise all.
  2. Figure 4: The resolution of this figure needs to be improved.
  3. Table 2: The sentences need more separation to be easy to follow.
  4. The conclusion: seems very short and needs to be more comprehensive.
  5. Line 170, the author wrote: In a study, they were severe enough that 27 of 302 patients had to be withdrawn from the trial [53]. Please add more details in the review about this study to explain its hypothesis.
  6. Line 185: Please add more details about faecal microbiota transplantation (FMT).
  7. All abbreviations should be defined at 1st mention, so please write the full name when first mentioned; then you can use the abbreviation later in the article. Moreover, the list of abbreviations at the end needs to be reformed.
  • Line 60, the author wrote: consumption are 2 major factors, please change (2) into (two).
  • Line 172, the author wrote: Anti-tumor necrosis factor-alpha (TNF-α), please correct its name.
  • Line 179, the author wrote: placebo group treating CD and UC, Please change it to: ulcerative colitis (UC) and Crohn’s disease (CD).
  • Line 243, the author wrote: high-performance liquid chromatography, please add its abbreviation.

Best regards

Author Response

After reviewing the review, I believe that it is generally well-written and structured. Moreover, the review is technically sound. Only a few comments are necessary. 

  1. The legends of the figures are very long and lack clarity, with some abbreviations left unexplained, which reduces the interpretability of the data. Please revise all. 

Response: We agree and have revised the figure legends that were unnecessarily long. All abbreviations have been reviewed and updated accordingly. Additionally, the abbreviations in each figure legend have been reordered alphabetically.

  1. Figure 4: The resolution of this figure needs to be improved. 

Response: We understand the concern and have re-exported the figure from our design software at the highest possible resolution and uploaded the revised version.

  1. Table 2: The sentences need more separation to be easy to follow. 

Response: We understand and have made the requested revisions. In addition, building on this comment, we have further refined Table 1 to make it more concise and easier to follow.

  1. The conclusion: seems very short and needs to be more comprehensive. 

Response: We agree and have made the revisions accordingly. Please refer to lines 61–81under Section 8 for the updated content.

  1. Line 170, the author wrote: In a study, they were severe enough that 27 of 302 patients had to be withdrawn from the trial [53]. Please add more details in the review about this study to explain its hypothesis. 

Response: We agree and have revised the text accordingly. Please refer to lines 166–171 for the revised content.

  1. Line 185: Please add more details about faecal microbiota transplantation (FMT). 

Response: We agree and have made the necessary revisions. Please refer to lines 186–190 and 194–195 for the revised content.

  1. All abbreviations should be defined at 1st mention, so please write the full name when first mentioned; then you can use the abbreviation later in the article. Moreover, the list of abbreviations at the end needs to be reformed. 

Response: We agree and have made the necessary revisions. Specifically, changes were made to ‘DOX’ (line 412) and ‘FDA’ (line 187) to ensure clarity and consistency.

  1. Line 60, the author wrote: consumption are 2 major factors, please change (2) into (two). 

Response: We agree with the suggestion and have made the revision at line 60.

  1. Line 172, the author wrote: Anti-tumor necrosis factor-alpha (TNF-α), please correct its name. 

Response: We agree and have revised both the name (line 172) and the corresponding entry in the abbreviation list (line 92).

  1. Line 179, the author wrote: placebo group treating CD and UC, Please change it to: ulcerative colitis (UC) and Crohn’s disease (CD). 

Response: We agree and have revised lines 179–180 accordingly.

  1. Line 243, the author wrote: high-performance liquid chromatography, please add its abbreviation. 

Response: We agree and have revised both line 250 and the corresponding entry in the abbreviation list (line 123).

Reviewer 2 Report

Comments and Suggestions for Authors

This is a timely and well-referenced review article that thoroughly discusses the emerging roles of nanomedicine in addressing gut dysbiosis, inflammatory bowel disease (IBD), and colorectal cancer (CRC). The manuscript is well-structured, comprehensive, and integrates both diagnostic and therapeutic perspectives. However, a few issues require attention to improve clarity, depth, and scientific rigor.

  1. Scope and Novelty:

    • While the review includes many nanomedical strategies, it lacks a critical comparison with existing conventional therapies in terms of limitations, delivery challenges, and patient outcomes. Please elaborate more explicitly on how nanotechnology overcomes specific bottlenecks in current IBD and CRC treatment.

  2. Mechanistic Depth:

    • In several sections (e.g., Sections 6.1 and 6.3), therapeutic mechanisms are described superficially. For example, you mention that “nanoparticles target inflamed tissues” but do not explain how targeting is achieved (e.g., via ligand–receptor interaction, pH, ROS, enzyme-responsive systems). Please expand the mechanistic explanation of targeting and release for key formulations.

  3. Clinical Translation:

    • There is limited discussion on how close these nano-formulations are to clinical application. Are any of the described technologies in clinical trials? Please include a short summary table or paragraph (perhaps in Section 7) listing nanomedicines currently in Phase I–III trials for IBD or CRC.

  4. Microbiome Modulation via Nanotechnology:

    • The section on nano-synbiotics is novel and important. However, the therapeutic outcomes (e.g., weight gain, colon length restoration) could be better contextualized. Please clarify whether these outcomes are primary endpoints in preclinical models and how they relate to clinical improvement in IBD/CRC.

  5. Safety and Toxicity Considerations:

    • The review would benefit from a dedicated paragraph or subsection discussing the toxicity, immunogenicity, and clearance pathways of nanomaterials. For instance, how do silver nanoparticles or carbon nanotubes behave in vivo long-term?

    • Figures:

      • Figures are visually informative but too dense. Consider simplifying labels or using color coding to differentiate diagnostic vs. therapeutic strategies.

        References:

        • Most references are recent and relevant. However, some highly cited foundational studies on microbiome dysbiosis and nanoparticle pharmacokinetics are missing. Consider including key reviews from Nature Nanotechnology, Gut, and Advanced Drug Delivery Reviews.

          Abbreviations:

          • Define all abbreviations at first use in the text (e.g., CSC, DDS, EPR). The abbreviation list in figures is helpful but should be standardized in the main text.

            Conclusion Section:

            • The conclusion could be more impactful if it included a roadmap for future research — e.g., the need for microbiome-specific delivery platforms, multi-omics integration with nanomedicine, or real-time diagnostic–therapeutic (theranostic) systems.

Comments on the Quality of English Language

Language and Grammar:

    • Overall English usage is strong, but certain sentences are long and repetitive. For example, lines 70–74 could be shortened and clarified. Consider professional editing to improve fluency.

Author Response

This is a timely and well-referenced review article that thoroughly discusses the emerging roles of nanomedicine in addressing gut dysbiosis, inflammatory bowel disease (IBD), and colorectal cancer (CRC). The manuscript is well-structured, comprehensive, and integrates both diagnostic and therapeutic perspectives. However, a few issues require attention to improve clarity, depth, and scientific rigor. 

  1. Scope and Novelty: 
  • While the review includes many nanomedical strategies, it lacks a critical comparison with existing conventional therapies in terms of limitations, delivery challenges, and patient outcomes. Please elaborate more explicitly on how nanotechnology overcomes specific bottlenecks in current IBD and CRC treatment. 

Response: We agree and have revised several paragraphs to better elaborate on the advantages of nanotechnology over current conventional therapies. The edited sections include lines 251–254, 293–295, 301, 302-303, 308–309, 316–318, 346–348, 354–359, 361, 369–372, 385–386, 403, 408–410, 418–419, 440–441, 446–448, 536–538, 559–561, 588–590, 604–606, 618–621, 629–631, 640–642, 664–665, 668–670, and 683–685.

  1. Mechanistic Depth: 
  • In several sections (e.g., Sections 6.1 and 6.3), therapeutic mechanisms are described superficially. For example, you mention that “nanoparticles target inflamed tissues” but do not explain how targeting is achieved (e.g., via ligand–receptor interaction, pH, ROS, enzyme-responsive systems). Please expand the mechanistic explanation of targeting and release for key formulations. 

Response: We agree and have revised the manuscript accordingly. Please refer to lines 354–359, 414–416, 460–463, 510, 570–574, and 680–681 for the updates.

  1. Clinical Translation: 
  • There is limited discussion on how close these nano-formulations are to clinical application. Are any of the described technologies in clinical trials? Please include a short summary table or paragraph (perhaps in Section 7) listing nanomedicines currently in Phase I–III trials for IBD or CRC. 

Response: We agree and have revised the manuscript to include the current status of clinical trials for relevant technologies. Please refer to lines 25–36 of Section 7.

  1. Microbiome Modulation via Nanotechnology: 
  • The section on nano-synbiotics is novel and important. However, the therapeutic outcomes (e.g., weight gain, colon length restoration) could be better contextualized. Please clarify whether these outcomes are primary endpoints in preclinical models and how they relate to clinical improvement in IBD/CRC. 

Response: We agree and have revised accordingly. Please refer to lines 570–574.

  1. Safety and Toxicity Considerations: 
  • The review would benefit from a dedicated paragraph or subsection discussing the toxicity, immunogenicity, and clearance pathways of nanomaterials. For instance, how do silver nanoparticles or carbon nanotubes behave in vivo long-term? 

Response: We agree and have revised the manuscript by including additional relevant references on the toxicity, clearance, and safety profile of nanotechnology, highlighting their relevance to IBD and CRC treatment. Please refer to lines 3–24 of Section 7.  

  1. Figures: 
  • Figures are visually informative but too dense. Consider simplifying labels or using color coding to differentiate diagnostic vs. therapeutic strategies. 

Response: We agree and have revised the labels to be as succinct as possible.

  1. References: 
  • Most references are recent and relevant. However, some highly cited foundational studies on microbiome dysbiosis and nanoparticle pharmacokinetics are missing. Consider including key reviews from Nature Nanotechnology, Gut, and Advanced Drug Delivery Reviews

Response: We agree and have revised the manuscript accordingly. Two references from Advanced Drug Delivery Reviews have been added:

  • Durán-Lobato M, López-Estévez AM, Cordeiro AS, Dacoba TG, Crecente-Campo J, Torres D, Alonso MJ. Nanotechnologies for the delivery of biologicals: Historical perspective and current landscape. Advanced Drug Delivery Reviews. 2021 Sep 1;176:113899.
  • Eftekharifar M, Heidari R, Mohaghegh N, Najafabadi AH, Heidari H. Advances in photoactivated carbon-based nanostructured materials for targeted cancer therapy. Advanced Drug Delivery Reviews. 2025 May 10:115604.
  1. Abbreviations: 
  • Define all abbreviations at first use in the text (e.g., CSC, DDS, EPR). The abbreviation list in figures is helpful but should be standardized in the main text. 

Response: We agree and have revised.

  1. Conclusion Section: 
  • The conclusion could be more impactful if it included a roadmap for future research — e.g., the need for microbiome-specific delivery platforms, multi-omics integration with nanomedicine, or real-time diagnostic–therapeutic (theranostic) systems. 

Response: We agree and have revised, please refer to lines 78-81 of Section 7.

  1. Language and Grammar: 
  • Overall English usage is strong, but certain sentences are long and repetitive. For example, lines 70–74 could be shortened and clarified. Consider professional editing to improve fluency. 

Response: We agree and have revised accordingly. The original lines 70–74 have been updated (now lines 69–76), and additional sections of the manuscript have also been added for greater clarity.